# TopRank: A Practical Algorithm for Online Stochastic Ranking

**Tor Lattimore**
DeepMind

**Branislav Kveton**
Google

**Shuai Li**
The Chinese University of Hong Kong

**Csaba Szepesvári**
DeepMind and University of Alberta

## Abstract

Online learning to rank is a sequential decision-making problem where in each round the learning agent chooses a list of items and receives feedback in the form of clicks from the user. Many sample-efficient algorithms have been proposed for this problem that assume a specific click model connecting rankings and user behavior. We propose a generalized click model that encompasses many existing models, including the position-based and cascade models. Our generalization motivates a novel online learning algorithm based on topological sort, which we call `TopRank`. `TopRank` is *(a)* more natural than existing algorithms, *(b)* has stronger regret guarantees than existing algorithms with comparable generality, *(c)* has a more insightful proof that leaves the door open to many generalizations, and *(d)* outperforms existing algorithms empirically.

## 1 Introduction

Learning to rank is an important problem with numerous applications in web search and recommender systems [11]. Broadly speaking, the goal is to learn an ordered list of $K$ items from a larger collection of size $L$ that maximizes the satisfaction of the user, often conditioned on a query. This problem has traditionally been studied in the offline setting, where the ranking policy is learned from manually-annotated relevance judgments. It has been observed that the feedback of users can be used to significantly improve existing ranking policies [1, 16]. This is the main motivation for online learning to rank, where the goal is to adaptively maximize the user satisfaction.

Numerous methods have been proposed for online learning to rank, both in the adversarial [12, 13] and stochastic settings. Our focus is on the stochastic setup where recent work has leveraged click models to mitigate the curse of dimensionality that arises from the combinatorial nature of the action-set. A click model is a model for how users click on items in rankings and is widely studied by the information retrieval community [2]. One popular click model in learning to rank is the cascade model (CM), which assumes that the user scans the ranking from top to bottom, clicking on the first item they find attractive [6, 3, 7, 18, 10, 5]. Another model is the position-based model (PBM), where the probability that the user clicks on an item depends on its position and attractiveness, but not on the surrounding items [8].

The cascade and position-based models have relatively few parameters, which is both a blessing and a curse. On the positive side, a small model is easy to learn. More negatively, there is a danger that a simplistic model will have a large approximation error. In fact, it has been observed experimentally that no single existing click model captures the behavior of an entire population of users [4]. Zoghi et al. [17] recently showed that under reasonable assumptions a single online learning algorithm can

learn the optimal list of items in a much larger class of click models that includes both the cascade and position-based models.

We build on the work of Zoghi et al. [17] and generalize it non-trivially in multiple directions. First, we propose a general model of user interaction where the problem of finding most attractive list can be posed as a sorting problem with noisy feedback. An interesting characteristic of our model is that the click probability does not factor into the examination probability of the position and the attractiveness of the item at that position. Second, we propose an online learning algorithm for finding the most attractive list, which we call TopRank. The key idea in the design of the algorithm is to maintain a partial order over the items that is refined as the algorithm observes more data. The new algorithm is simultaneously simpler, more principled and empirically outperforms the algorithm of Zoghi et al. [17]. We also provide an analysis of the cumulative regret of TopRank that is simple, insightful and strengthens the results by Zoghi et al. [17], despite the weaker assumptions.

## 2  Online learning to rank

We assume the total numbers of items $L$ is larger than the number of available slots $K$ and that the collection of items is $[L] = \{1, 2, \ldots, L\}$. A permutation on finite set $X$ is an invertible function $\sigma : X \to X$ and the set of all permutations on $X$ is denoted by $\Pi(X)$. The set of actions $\mathcal{A}$ is the set of permutations $\Pi([L])$, where for each $a \in \mathcal{A}$ the value $a(k)$ should be interpreted as the identity of the item placed at the $k$th position. Equivalently, item $i$ is placed at position $a^{-1}(i)$. The user does not observe items in positions $k > K$ so the order of $a(k+1), \ldots, a(L)$ is not important and is included only for notational convenience. We adopt the convention throughout that $i$ and $j$ represent items while $k$ represents a position.

The online ranking problem proceeds over $n$ rounds. In each round $t$ the learner chooses an action $A_t \in \mathcal{A}$ based on its observations so far and observes binary random variables $C_{t1}, \ldots, C_{tL}$ where $C_{ti} = 1$ if the user clicked on item $i$. We assume a stochastic model where the probability that the user clicks on position $k$ in round $t$ only depends on $A_t$ and is given by

$$\mathbb{P}(C_{tA_t(k)} = 1 \mid A_t = a) = v(a, k)$$

with $v : \mathcal{A} \times [L] \to [0, 1]$ an unknown function. Another way of writing this is that the conditional probability that the user clicks on item $i$ in round $t$ is $\mathbb{P}(C_{ti} = 1 \mid A_t = a) = v(a, a^{-1}(i))$.

The performance of the learner is measured by the expected cumulative regret, which is the deficit suffered by the learner relative to the omniscient strategy that knows the optimal ranking in advance.

$$R_n = n \max_{a \in \mathcal{A}} \sum_{k=1}^{K} v(a, k) - \mathbb{E} \left[ \sum_{t=1}^{n} \sum_{i=1}^{L} C_{ti} \right] = \max_{a \in \mathcal{A}} \mathbb{E} \left[ \sum_{t=1}^{n} \sum_{k=1}^{K} (v(a, k) - v(A_t, k)) \right].$$

**Remark 1.** *We do not assume that $C_{t1}, \ldots, C_{tL}$ are independent or that the user can only click on one item.*

## 3  Modeling assumptions

In previous work on online learning to rank it was assumed that $v$ factors into $v(a, k) = \alpha(a(k))\chi(a, k)$ where $\alpha : [L] \to [0, 1]$ is the attractiveness function and $\chi(a, k)$ is the probability that the user examines position $k$ given ranking $a$. Further restrictions are made on the examination function $\chi$. For example, in the document-based model it is assumed that $\chi(a, k) = \mathbb{1}\{k \leq K\}$. In this work we depart from this standard by making assumptions directly on $v$. The assumptions are sufficiently relaxed that the model subsumes the document-based, position-based and cascade models, as well as the factored model studied by Zoghi et al. [17]. See the supplementary material for a proof of this. Our first assumption uncontroversially states that the user does not click on items they cannot see.

**Assumption 1.** $v(a, k) = 0$ *for all* $k > K$.

Although we do not assume an explicit factorization of the click probability into attractiveness and examination functions, we do assume there exists an unknown attractiveness function $\alpha : [L] \to [0, 1]$ that satisfies the following assumptions. In all classical click models the optimal ranking is to sort the

## Algorithm 1 TopRank

1:  $G_1 \leftarrow \emptyset$ and $c \leftarrow \frac{4\sqrt{2/\pi}}{\mathrm{erf}(\sqrt{2})}$
2:  **for** $t = 1, \ldots, n$ **do**
3:      $d \leftarrow 0$
4:      **while** $[L] \setminus \bigcup_{c=1}^{d} \mathcal{P}_{tc} \neq \emptyset$ **do**
5:          $d \leftarrow d + 1$
6:          $\mathcal{P}_{td} \leftarrow \min_{G_t}\left([L] \setminus \bigcup_{c=1}^{d-1} \mathcal{P}_{tc}\right)$
7:      Choose $A_t$ uniformly at random from $\mathcal{A}(\mathcal{P}_{t1}, \ldots, \mathcal{P}_{td})$
8:      Observe click indicators $C_{ti} \in \{0, 1\}$ for all $i \in [L]$
9:      **for all** $(i, j) \in [L]^2$ **do**
10:          $U_{tij} \leftarrow \begin{cases} C_{ti} - C_{tj} & \text{if } i, j \in \mathcal{P}_{td} \text{ for some } d \\ 0 & \text{otherwise} \end{cases}$
11:          $S_{tij} \leftarrow \sum_{s=1}^{t} U_{sij}$ and $N_{tij} \leftarrow \sum_{s=1}^{t} |U_{sij}|$
12:      $G_{t+1} \leftarrow G_t \cup \left\{(j, i) : S_{tij} \geq \sqrt{2 N_{tij} \log\left(\frac{c}{\delta} \sqrt{N_{tij}}\right)} \text{ and } N_{tij} > 0\right\}$

items in order of decreasing attractiveness. Rather than deriving this from other assumptions, we will simply assume that $v$ satisfies this criteria. We call action $a$ optimal if $\alpha(a(k)) = \max_{k' \geq k} \alpha(a(k'))$ for all $k \in [K]$. The optimal action need not be unique if $\alpha$ is not injective, but the sequence $\alpha(a(1)), \ldots, \alpha(a(K))$ is the same for all optimal actions.

**Assumption 2.** *Let* $a^* \in \mathcal{A}$ *be an optimal action. Then* $\max_{a \in \mathcal{A}} \sum_{k=1}^{K} v(a, k) = \sum_{k=1}^{K} v(a^*, k)$.

The next assumption asserts that if $a$ is an action and $i$ is more attractive than $j$, then exchanging the positions of $i$ and $j$ can only decrease the likelihood of clicking on the item in slot $a^{-1}(i)$. Fig. 1 illustrates the two cases. The probability of clicking on the second position is larger in $a$ than in $a'$. On the other hand, the probability of clicking on the fourth position is larger in $a'$ than in $a$.

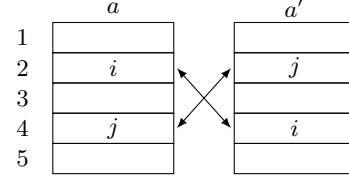

than in $a$. The assumption is actually slightly stronger than this because it also specifies a lower bound on the amount by which one probability is larger than another in terms of the attractiveness function.

Figure 1: The probability of clicking on the second position is larger in $a$ than $a'$. The pattern reverses for the fourth position.

**Assumption 3.** *Let $i$ and $j$ be items with $\alpha(i) \geq \alpha(j)$ and let $\sigma : \mathcal{A} \to \mathcal{A}$ be the permutation that exchanges $i$ and $j$ and leaves other items unchanged. Then for any action $a \in \mathcal{A}$,*

$$v(a, a^{-1}(i)) \geq \frac{\alpha(i)}{\alpha(j)} v(\sigma \circ a, a^{-1}(i)).$$

Our final assumption is that for any action $a$ with $\alpha(a(k)) = \alpha(a^*(k))$ the probability of clicking on the $k$th position is at least as high as the probability of clicking on the $k$th position for the optimal action. This assumption makes sense if the user is scanning the items from the first position until the last, clicking on items they find attractive until some level of satisfaction is reached. Under this assumption the user is least likely to examine position $k$ under the optimal ranking.

**Assumption 4.** *For any action $a$ and optimal action $a^*$ with $\alpha(a(k)) = \alpha(a^*(k))$ it holds that $v(a, k) \geq v(a^*, k)$.*

## 4 Algorithm

Before we present our algorithm, we introduce some basic notation. Given a relation $G \subseteq [L]^2$ and $X \subseteq [L]$, let $\min_G(X) = \{i \in X : (i, j) \notin G \text{ for all } j \in X\}$. When $X$ is nonempty and $G$ does not have cycles, then $\min_G(X)$ is nonempty. Let $\mathcal{P}_1, \ldots, \mathcal{P}_d$ be a *partition* of $[L]$ so that $\cup_{c \leq d} \mathcal{P}_c = [L]$ and $\mathcal{P}_c \cap \mathcal{P}_{c'} = \emptyset$ for any $c \neq c'$. We refer to each subset in the partition, $\mathcal{P}_c$ for $c \leq d$, as a *block*. Let $\mathcal{A}(\mathcal{P}_1, \ldots, \mathcal{P}_d)$ be the set of actions $a$ where the items in $\mathcal{P}_1$ are placed at the first $|\mathcal{P}_1|$ positions,

the items in $\mathcal{P}_2$ are placed at the next $|\mathcal{P}_2|$ positions, and so on. Specifically,

$$\mathcal{A}(\mathcal{P}_1, \ldots, \mathcal{P}_d) = \left\{ a \in \mathcal{A} : \max_{i \in \mathcal{P}_c} a^{-1}(i) \leq \min_{i \in \mathcal{P}_{c+1}} a^{-1}(i) \text{ for all } c \in [d-1] \right\} .$$

Our algorithm is presented in Algorithm 1. We call it `TopRank`, because it maintains a *topological order* of items in each round. The order is represented by relation $G_t$, where $G_1 = \emptyset$. In each round, `TopRank` computes a partition of $[L]$ by iteratively peeling off minimum items according to $G_t$. Then it randomizes items in each block of the partition and maintains statistics on the relative number of clicks between pairs of items in the same block. A pair of items $(j, i)$ is added to the relation once item $i$ receives sufficiently more clicks than item $j$ during rounds where the items are in the same block. The reader should interpret $(j, i) \in G_t$ as meaning that `TopRank` collected enough evidence up to round $t$ to conclude that $\alpha(j) < \alpha(i)$.

**Remark 2.** *The astute reader will notice that the algorithm is not well defined if $G_t$ contains cycles. The analysis works by proving that this occurs with low probability and the behavior of the algorithm may be defined arbitrarily whenever a cycle is encountered. Assumption 1 means that items in position $k > K$ are never clicked. As a consequence, the algorithm never needs to actually compute the blocks $\mathcal{P}_{td}$ where $\min \mathcal{I}_{td} > K$ because items in these blocks are never shown to the user.*

Shortly we give an illustration of the algorithm, but first introduce the notation to be used in the analysis. Let $\mathcal{I}_{td}$ be the slots of the ranking where items in $\mathcal{P}_{td}$ are placed,

$$\mathcal{I}_{td} = [|\cup_{c \leq d} \mathcal{P}_{tc}|] \setminus [|\cup_{c < d} \mathcal{P}_{tc}|] .$$

Furthermore, let $D_{ti}$ be the block with item $i$, so that $i \in \mathcal{P}_{tD_{ti}}$. Let $M_t = \max_{i \in [L]} D_{ti}$ be the number of blocks in the partition in round $t$.

**Illustration**  Suppose $L = 5$ and $K = 4$ and in round $t$ the relation is $G_t = \{(3,1), (5,2), (5,3)\}$. This indicates the algorithm has collected enough data to believe that item 3 is less attractive than item 1 and that item 5 is less attractive than items 2 and 3. The relation is depicted in Fig. 2 where an arrow from $j$ to $i$ means that $(j, i) \in G_t$. In round $t$ the first three positions in the ranking will contain items from $\mathcal{P}_{t1} = \{1, 2, 4\}$, but with random order. The fourth position will be item 3 and item 5 is not shown to the user. Note that $M_t = 3$ here and $D_{t2} = 1$ and $D_{t5} = 3$.

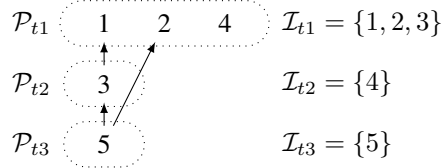

$\mathcal{P}_{t1}$  1  2  4   $\mathcal{I}_{t1} = \{1, 2, 3\}$

$\mathcal{P}_{t2}$  3   $\mathcal{I}_{t2} = \{4\}$

$\mathcal{P}_{t3}$  5   $\mathcal{I}_{t3} = \{5\}$

Figure 2: Illustration of partition produced by topological sort

**Remark 3.** `TopRank` *is not an elimination algorithm. In the scenario described above, item 5 is not shown to the user, but it could happen that later $(4, 2)$ and $(4, 3)$ are added to the relation and then* `TopRank` *will start randomizing between items 4 and 5 for the fourth position.*

## 5 Regret analysis

**Theorem 1.** *Let function $v$ satisfy Assumptions 1–4 and $\alpha(1) > \alpha(2) > \cdots > \alpha(L)$. Let $\Delta_{ij} = \alpha(i) - \alpha(j)$ and $\delta \in (0, 1)$. Then the $n$-step regret of* `TopRank` *is bounded from above as*

$$R_n \leq \delta n K L^2 + \sum_{j=1}^{L} \sum_{i=1}^{\min\{K, j-1\}} \left( 1 + \frac{6(\alpha(i) + \alpha(j)) \log\left(\frac{c\sqrt{n}}{\delta}\right)}{\Delta_{ij}} \right) .$$

*Furthermore, $R_n \leq \delta n K L^2 + KL + \sqrt{4K^3 Ln \log\left(\frac{c\sqrt{n}}{\delta}\right)}$.*

By choosing $\delta = n^{-1}$ the theorem shows that the expected regret is at most

$$R_n = O\left( \sum_{j=1}^{L} \sum_{i=1}^{\min\{K, j-1\}} \frac{\alpha(i) \log(n)}{\Delta_{ij}} \right) \qquad \text{and} \qquad R_n = O\left( \sqrt{K^3 Ln \log(n)} \right) .$$

The algorithm does not make use of any assumed ordering on $\alpha(\cdot)$, so the assumption is only used to allow for a simple expression for the regret. The only algorithm that operates under comparably general assumptions is `BatchRank` for which the problem-dependent regret is a factor of $K^2$ worse and the dependence on the suboptimality gap is replaced by a dependence on the minimal suboptimality gap.

The core idea of the proof is to show that *(a)* if the algorithm is suffering regret as a consequence of misplacing an item, then it is gaining information about the relation of the items so that $G_t$ will gain elements and *(b)* once $G_t$ is sufficiently rich the algorithm is playing optimally. Let $\mathcal{F}_t = \sigma(A_1, C_1, \ldots, A_t, C_t)$ and $\mathbb{P}_t(\cdot) = \mathbb{P}(\cdot \mid \mathcal{F}_t)$ and $\mathbb{E}_t[\cdot] = \mathbb{E}[\cdot \mid \mathcal{F}_t]$. For each $t \in [n]$ let $F_t$ to be the failure event that there exists $i \neq j \in [L]$ and $s < t$ such that $N_{sij} > 0$ and

$$\left| S_{sij} - \sum_{u=1}^{s} \mathbb{E}_{u-1}\left[U_{uij} \mid U_{uij} \neq 0\right] |U_{uij}| \right| \geq \sqrt{2N_{sij}\log(c\sqrt{N_{sij}}/\delta)}.$$

**Lemma 1.** *Let $i$ and $j$ satisfy $\alpha(i) \geq \alpha(j)$ and $d \geq 1$. On the event that $i, j \in \mathcal{P}_{sd}$ and $d \in [M_s]$ and $U_{sij} \neq 0$, the following hold almost surely:*

*(a)* $\mathbb{E}_{s-1}[U_{sij} \mid U_{sij} \neq 0] \geq \dfrac{\Delta_{ij}}{\alpha(i) + \alpha(j)}$        *(b)* $\mathbb{E}_{s-1}[U_{sji} \mid U_{sji} \neq 0] \leq 0$.

*Proof.* For the remainder of the proof we focus on the event that $i, j \in \mathcal{P}_{sd}$ and $d \in [M_s]$ and $U_{sij} \neq 0$. We also discard the measure zero subset of this event where $\mathbb{P}_{s-1}(U_{sij} \neq 0) = 0$. From now on we omit the 'almost surely' qualification on conditional expectations. Under these circumstances the definition of conditional expectation shows that

$$\mathbb{E}_{s-1}[U_{sij} \mid U_{sij} \neq 0] = \frac{\mathbb{P}_{s-1}(C_{si} = 1, C_{sj} = 0) - \mathbb{P}_{s-1}(C_{si} = 0, C_{sj} = 1)}{\mathbb{P}_{s-1}(C_{si} \neq C_{sj})}$$

$$= \frac{\mathbb{P}_{s-1}(C_{si} = 1) - \mathbb{P}_{s-1}(C_{sj} = 1)}{\mathbb{P}_{s-1}(C_{si} \neq C_{sj})} \geq \frac{\mathbb{P}_{s-1}(C_{si} = 1) - \mathbb{P}_{s-1}(C_{sj} = 1)}{\mathbb{P}_{s-1}(C_{si} = 1) + \mathbb{P}_{s-1}(C_{sj} = 1)}$$

$$= \frac{\mathbb{E}_{s-1}[v(A_s, A_s^{-1}(i)) - v(A_s, A_s^{-1}(j))]}{\mathbb{E}_{s-1}[v(A_s, A_s^{-1}(i)) + v(A_s, A_s^{-1}(j))]}, \tag{1}$$

where in the second equality we added and subtracted $\mathbb{P}_{s-1}(C_{si} = 1, C_{sj} = 1)$. By the design of `TopRank`, the items in $\mathcal{P}_{td}$ are placed into slots $\mathcal{I}_{td}$ uniformly at random. Let $\sigma$ be the permutation that exchanges the positions of items $i$ and $j$. Then using Assumption 3,

$$\mathbb{E}_{s-1}[v(A_s, A_s^{-1}(i))] = \sum_{a \in \mathcal{A}} \mathbb{P}_{s-1}(A_s = a)v(a, a^{-1}(i)) \geq \frac{\alpha(i)}{\alpha(j)} \sum_{a \in \mathcal{A}} \mathbb{P}_{s-1}(A_s = a)v(\sigma \circ a, a^{-1}(i))$$

$$= \frac{\alpha(i)}{\alpha(j)} \sum_{a \in \mathcal{A}} \mathbb{P}_{s-1}(A_s = \sigma \circ a)v(\sigma \circ a, (\sigma \circ a)^{-1}(j)) = \frac{\alpha(i)}{\alpha(j)} \mathbb{E}_{s-1}[v(A_s, A_s^{-1}(j))],$$

where the second equality follows from the fact that $a^{-1}(i) = (\sigma \circ a)^{-1}(j)$ and the definition of the algorithm ensuring that $\mathbb{P}_{s-1}(A_s = a) = \mathbb{P}_{s-1}(A_s = \sigma \circ a)$. The last equality follows from the fact that $\sigma$ is a bijection. Using this and continuing the calculation in Eq. (1) shows that

$$\frac{\mathbb{E}_{s-1}\left[v(A_s, A_s^{-1}(i)) - v(A_s, A_s^{-1}(j))\right]}{\mathbb{E}_{s-1}\left[v(A_s, A_s^{-1}(i)) + v(A_s, A_s^{-1}(j))\right]} = 1 - \frac{2}{1 + \mathbb{E}_{s-1}\left[v(A_s, A_s^{-1}(i))\right]/\mathbb{E}_{s-1}\left[v(A_s, A_s^{-1}(j))\right]}$$

$$\geq 1 - \frac{2}{1 + \alpha(i)/\alpha(j)} = \frac{\alpha(i) - \alpha(j)}{\alpha(i) + \alpha(j)} = \frac{\Delta_{ij}}{\alpha(i) + \alpha(j)}.$$

The second part follows from the first since $U_{sji} = -U_{sij}$.    $\square$

The next lemma shows that the failure event occurs with low probability.

**Lemma 2.** *It holds that $\mathbb{P}(F_n) \leq \delta L^2$.*

*Proof.* The proof follows immediately from Lemma 1, the definition of $F_n$, the union bound over all pairs of actions, and a modification of the Azuma-Hoeffding inequality in Lemma 6.    $\square$

**Lemma 3.** *On the event $F_t^c$ it holds that $(i, j) \notin G_t$ for all $i < j$.*

*Proof.* Let $i < j$ so that $\alpha(i) \geq \alpha(j)$. On the event $F_t^c$ either $N_{sji} = 0$ or

$$S_{sji} - \sum_{u=1}^{s} \mathbb{E}_{u-1}[U_{uji} \mid U_{uji} \neq 0]|U_{uji}| < \sqrt{2N_{sji} \log\left(\frac{c}{\delta}\sqrt{N_{sji}}\right)} \qquad \text{for all } s < t.$$

When $i$ and $j$ are in different blocks in round $u < t$, then $U_{uji} = 0$ by definition. On the other hand, when $i$ and $j$ are in the same block, $\mathbb{E}_{u-1}[U_{uji} \mid U_{uji} \neq 0] \leq 0$ almost surely by Lemma 1. Based on these observations,

$$S_{sji} < \sqrt{2N_{sji} \log\left(\frac{c}{\delta}\sqrt{N_{sji}}\right)} \qquad \text{for all } s < t,$$

which by the design of TopRank implies that $(i, j) \notin G_t$. □

**Lemma 4.** *Let $I_{td}^* = \min \mathcal{P}_{td}$ be the most attractive item in $\mathcal{P}_{td}$. Then on event $F_t^c$, it holds that $I_{td}^* \leq 1 + \sum_{c<d} |\mathcal{P}_{td}|$ for all $d \in [M_t]$.*

*Proof.* Let $i^* = \min \cup_{c \geq d} \mathcal{P}_{tc}$. Then $i^* \leq 1 + \sum_{c<d} |\mathcal{P}_{td}|$ holds trivially for any $\mathcal{P}_{t1}, \ldots, \mathcal{P}_{tM_t}$ and $d \in [M_t]$. Now consider two cases. Suppose that $i^* \in \mathcal{P}_{td}$. Then it must be true that $i^* = I_{td}^*$ and our claim holds. On other hand, suppose that $i^* \in \mathcal{P}_{tc}$ for some $c > d$. Then by Lemma 3 and the design of the partition, there must exist a sequence of items $i_d, \ldots, i_c$ in blocks $\mathcal{P}_{td}, \ldots, \mathcal{P}_{tc}$ such that $i_d < \cdots < i_c = i^*$. From the definition of $I_{td}^*$, $I_{td}^* \leq i_d < i^*$. This concludes our proof. □

**Lemma 5.** *On the event $F_n^c$ and for all $i < j$ it holds that $S_{nij} \leq 1 + \dfrac{6(\alpha(i) + \alpha(j))}{\Delta_{ij}} \log\left(\dfrac{c\sqrt{n}}{\delta}\right)$.*

*Proof.* The result is trivial when $N_{nij} = 0$. Assume from now on that $N_{nij} > 0$. By the definition of the algorithm arms $i$ and $j$ are not in the same block once $S_{tij}$ grows too large relative to $N_{tij}$, which means that

$$S_{nij} \leq 1 + \sqrt{2N_{nij} \log\left(\frac{c}{\delta}\sqrt{N_{nij}}\right)}.$$

On the event $F_n^c$ and part (a) of Lemma 1 it also follows that

$$S_{nij} \geq \frac{\Delta_{ij} N_{nij}}{\alpha(i) + \alpha(j)} - \sqrt{2N_{nij} \log\left(\frac{c}{\delta}\sqrt{N_{nij}}\right)}.$$

Combining the previous two displays shows that

$$\frac{\Delta_{ij} N_{nij}}{\alpha(i) + \alpha(j)} - \sqrt{2N_{nij} \log\left(\frac{c}{\delta}\sqrt{N_{nij}}\right)} \leq S_{nij} \leq 1 + \sqrt{2N_{nij} \log\left(\frac{c}{\delta}\sqrt{N_{nij}}\right)}$$

$$\leq (1 + \sqrt{2})\sqrt{N_{nij} \log\left(\frac{c}{\delta}\sqrt{N_{nij}}\right)}. \qquad (2)$$

Using the fact that $N_{nij} \leq n$ and rearranging the terms in the previous display shows that

$$N_{nij} \leq \frac{(1 + 2\sqrt{2})^2 (\alpha(i) + \alpha(j))^2}{\Delta_{ij}^2} \log\left(\frac{c\sqrt{n}}{\delta}\right).$$

The result is completed by substituting this into Eq. (2). □

*Proof of Theorem 1.* The first step in the proof is an upper bound on the expected number of clicks in the optimal list $a^*$. Fix time $t$, block $\mathcal{P}_{td}$, and recall that $I_{td}^* = \min \mathcal{P}_{td}$ is the most attractive item in $\mathcal{P}_{td}$. Let $k = A_t^{-1}(I_{td}^*)$ be the position of item $I_{td}^*$ and $\sigma$ be the permutation that exchanges items $k$ and $I_{td}^*$. By Lemma 4, $I_{td}^* \leq k$; and then from Assumptions 3 and 4, we have that

$v(A_t, k) \geq v(\sigma \circ A_t, k) \geq v(a^*, k)$. Based on this result, the expected number of clicks on $I^*_{td}$ is bounded from below by those on items in $a^*$,

$$\mathbb{E}_{t-1}\left[C_{tI^*_{td}}\right] = \sum_{k \in \mathcal{I}_{td}} \mathbb{P}_{t-1}(A_t^{-1}(I^*_{td}) = k)\mathbb{E}_{t-1}[v(A_t, k) \mid A_t^{-1}(I^*_{td}) = k]$$

$$= \frac{1}{|\mathcal{I}_{td}|} \sum_{k \in \mathcal{I}_{td}} \mathbb{E}_{t-1}[v(A_t, k) \mid A_t^{-1}(I^*_{td}) = k] \geq \frac{1}{|\mathcal{I}_{td}|} \sum_{k \in \mathcal{I}_{td}} v(a^*, k),$$

where we also used the fact that TopRank randomizes within each block to guarantee that $\mathbb{P}_{t-1}(A_t^{-1}(I^*_{td}) = k) = 1/|\mathcal{I}_{td}|$ for any $k \in \mathcal{I}_{td}$. Using this and the design of TopRank,

$$\sum_{k=1}^{K} v(a^*, k) = \sum_{d=1}^{M_t} \sum_{k \in \mathcal{I}_{td}} v(a^*, k) \leq \sum_{d=1}^{M_t} |\mathcal{I}_{td}|\mathbb{E}_{t-1}\left[C_{tI^*_{td}}\right].$$

Therefore, under event $F_t^c$, the conditional expected regret in round $t$ is bounded by

$$\sum_{k=1}^{K} v(a^*, k) - \mathbb{E}_{t-1}\left[\sum_{j=1}^{L} C_{tj}\right] \leq \mathbb{E}_{t-1}\left[\sum_{d=1}^{M_t} |\mathcal{P}_{td}|C_{tI^*_{td}} - \sum_{j=1}^{L} C_{tj}\right]$$

$$= \mathbb{E}_{t-1}\left[\sum_{d=1}^{M_t} \sum_{j \in \mathcal{P}_{td}} (C_{tI^*_{td}} - C_{tj})\right] = \sum_{d=1}^{M_t} \sum_{j \in \mathcal{P}_{td}} \mathbb{E}_{t-1}[U_{tI^*_{td}j}] \leq \sum_{j=1}^{L} \sum_{i=1}^{\min\{K,j-1\}} \mathbb{E}_{t-1}[U_{tij}]. \quad (3)$$

The last inequality follows by noting that $\mathbb{E}_{t-1}[U_{tI^*_{td}j}] \leq \sum_{i=1}^{\min\{K,j-1\}} \mathbb{E}_{t-1}[U_{tij}]$. To see this use part (a) of Lemma 1 to show that $\mathbb{E}_{t-1}[U_{tij}] \geq 0$ for $i < j$ and Lemma 4 to show that when $I^*_{td} > K$, then neither $I^*_{td}$ nor $j$ are not shown to the user in round $t$ so that $U_{tI^*_{td}j} = 0$. Substituting the bound in Eq. (3) into the regret leads to

$$R_n \leq nK\mathbb{P}(F_n) + \sum_{j=1}^{L} \sum_{i=1}^{\min\{K,j-1\}} \mathbb{E}\left[\mathbb{1}\{F_n^c\} S_{nij}\right], \quad (4)$$

where we used the fact that the maximum number of clicks over $n$ rounds is $nK$. The proof of the first part is completed by using Lemma 2 to bound the first term and Lemma 5 to bound the second. The problem independent bound follows from Eq. (4) and by stopping early in the proof of Lemma 5. The details are given in the supplementary material. □

**Lemma 6.** *Let $(\mathcal{F}_t)_{t=0}^n$ be a filtration and $X_1, X_2, \ldots, X_n$ be a sequence of $\mathcal{F}_t$-adapted random variables with $X_t \in \{-1, 0, 1\}$ and $\mu_t = \mathbb{E}[X_t \mid \mathcal{F}_{t-1}, X_t \neq 0]$. Then with $S_t = \sum_{s=1}^t (X_s - \mu_s|X_s|)$ and $N_t = \sum_{s=1}^t |X_s|$,*

$$\mathbb{P}\left(\text{exists } t \leq n : |S_t| \geq \sqrt{2N_t \log\left(\frac{c\sqrt{N_t}}{\delta}\right)} \text{ and } N_t > 0\right) \leq \delta, \quad \text{where } c = \frac{4\sqrt{2/\pi}}{\text{erf}(\sqrt{2})} \approx 3.43.$$

See the supplementary material for the proof.

We also provide a minimax lower bound, the proof of which is deferred to the supplementary material.

**Theorem 2.** *Suppose that $L = NK$ with $N$ an integer and $n \geq K$ and $n \geq N$ and $N \geq 8$. Then for any algorithm there exists a ranking problem such that $\mathbb{E}[R_n] \geq \sqrt{KLn}/(16\sqrt{2})$.*

The proof of this result only makes use of ranking problems in the document-based model. This also corresponds to a lower bound for $m$-sets in online linear optimization with semi-bandit feedback. Despite the simple setup and abundant literature, we are not aware of any work where a lower bound of this form is presented for this unstructured setting.

# 6 Experiments

We experiment with the *Yandex* dataset [15], a dataset of 167 million search queries. In each query, the user is shown 10 documents at positions 1 to 10 and the search engine records the clicks of the user. We select 60 frequent search queries from this dataset, and learn their CMs and PBMs using PyClick [2]. The parameters of the models are learned by maximizing the likelihood of observed clicks. Our goal is to rerank $L = 10$ most attractive items with the objective of maximizing the expected number of clicks at the first $K = 5$ positions. This is the same experimental setup as in Zoghi *et al.* [17]. This is a realistic scenario where the learning agent can only rerank highly attractive items that are suggested by some production ranker [16].

`TopRank` is compared to `BatchRank` [17] and `CascadeKL-UCB` [6]. We used the implementation of `BatchRank` by Zoghi *et al.* [17]. We do not compare to ranked bandits [12], because they have already been shown to perform poorly in stochastic click models, for instance by Zoghi *et al.* [17] and Katariya *et al.* [5]. The parameter $\delta$ in `TopRank` is set as $\delta = 1/n$, as suggested in Theorem 1. Fig. 3 illustrates

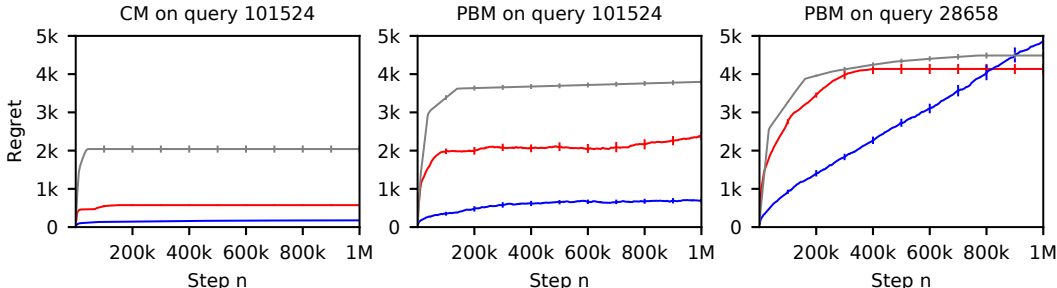

Figure 3: The $n$-step regret of `TopRank` (red), `CascadeKL-UCB` (blue), and `BatchRank` (gray) in three problems. The results are averaged over 10 runs. The error bars are the standard errors of our regret estimates.

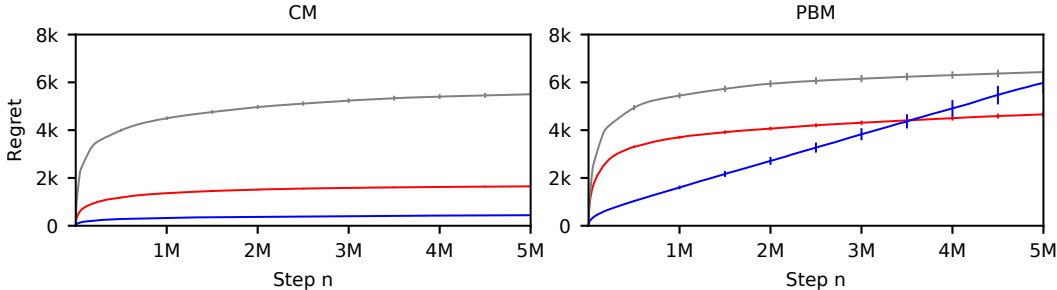

Figure 4: The $n$-step regret of `TopRank` (red), `CascadeKL-UCB` (blue), and `BatchRank` (gray) in two click models. The results are averaged over 60 queries and 10 runs per query. The error bars are the standard errors of our regret estimates.

the general trend on specific queries. In the cascade model, `CascadeKL-UCB` outperforms `TopRank`. This should not come as a surprise because `CascadeKL-UCB` heavily exploits the knowledge of the model. Despite being a more general algorithm, `TopRank` consistently outperforms `BatchRank` in the cascade model. In the position-based model, `CascadeKL-UCB` learns very good policies in about two thirds of queries, but suffers linear regret for the rest. In many of these queries, `TopRank` outperforms `CascadeKL-UCB` in as few as one million steps. In the position-based model, `TopRank` typically outperforms `BatchRank`.

The average regret over all queries is reported in Fig. 4. We observe similar trends to those in Fig. 3. In the cascade model, the regret of `CascadeKL-UCB` is about three times lower than that of `TopRank`, which is about three times lower than that of `BatchRank`. In the position-based model, the regret of `CascadeKL-UCB` is higher than that of `TopRank` after 4 million steps. The regret of `TopRank` is about 30% lower than that of `BatchRank`. In summary, we observe that `TopRank` improves over `BatchRank` in both the cascade and position-based models. The worse performance of `TopRank` relative to `CascadeKL-UCB` in the cascade model is offset by its robustness to multiple click models.

# 7  Conclusions

We introduced a new click model for online ranking that subsumes previous models. Despite the increased generality, the new algorithm enjoys stronger regret guarantees, an easier and more insightful proof and improved empirical performance. We hope the simplifications can inspire even more interest in online ranking. We also proved a lower bound for combinatorial linear semi-bandits with $m$-sets that improves on the bound by Uchiya et al. [14]. We do not currently have matching upper and lower bounds. The key to understanding minimax lower bounds is to identify what makes a problem hard. In many bandit models there is limited flexibility, but our assumptions are so weak that the space of all $v$ satisfying Assumptions 1–4 is quite large and we do not yet know what is the hardest case. This difficulty is perhaps even greater if the objective is to prove instance-dependent or asymptotic bounds where the results usually depend on solving a regret/information optimization problem [9]. Ranking becomes increasingly difficult as the number of items grows. In most cases where $L$ is large, however, one would expect the items to be structured and this should then be exploited. This has been done for the cascade model by assuming a linear structure [18, 10]. Investigating this possibility, but with more relaxed assumptions seems like an interesting future direction.

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
