[Reviews · NeurIPS 2018]

Reviewer 1



This paper tackles the problem of learning to rank items in the online setting. This paper proposes a new algorithm that uses confidence intervals on items preferences ordering in order to decide which item to assign to each position. Results show that the proposed approach empirically outperforms the current state-of-the-art in the re-ranking setting. The paper also provides an upper bound on the regret for the proposed approach, and a lower bound on the regret in ranking problems. Quality: I found the paper of overall good quality. Considering that there is some space left at the end, the paper could benefit from some more details to backup its claims: The abstract (line 8) claims that the proposed approach is "more natural than existing algorithms". Though I agree that this approach feels natural, other models are not discussed here. This makes it difficult to judge of how natural (or not) they are compared in comparison. The abstract (lines 8-9) and conclusion (line 241) claim that the proposed algorithm "enjoys stronger regret guarantees" than existing models, but the bounds of other algorithms are not reported nor discussed. How does the theoretical result of Thm. 1 compare with the bounds of other models (e.g. CascadeUCB, BatchRank)? Clarity: The paper reads well. It would be good to remind the reader what are K and L at the beginning of Sec. 2. Figures used to illustrate the model (in Sec. 3) and the algorithm (in Sec. 4) do not have a caption. I understand that this was meant to save space and/or alleviate the general presentation but there is empty space at the end of the paper. Nitpicking: - Reference [22] is a duplicate of [21]. Originality: This work addresses an existing problem with a novel approach. The proposed algorithm is intuitive and compares really well to the state of art. The consfidence intervals in Lem. 6 hold simultaneously for all times, which could be interesting for further analysis. Thm. 2 appears to be the first lower bound obtained by unifying different existing results. Significance: While the paper does not address a new problem, the learning to (re-)rank problem is still of interest for many people. The proposed algorithm empirically outperforms existing approaches, it would become one of the state-of-the-art baselines to consider. However, the significance of its regret upper bound (Thm. 1) is hard to evaluate without comparison with similar results of existing models. Thm. 2, on the other hand, might be the only current regret lower bound instance in the ranking problem.

Reviewer 2



Summary of the paper: A topological sort based algorithm named TopRank is proposed to deal with the online learning to rank problem, which does not rely on a factored value function model. TopRank maintains a graph of preference during learning, and chooses the ranking based on the topological ordering. The graph is updated by click results. Both regret bounds and empirical results on real data are provided to justify the effectiveness. Comments: This is a well written paper, which gives very clear exposition on modeling assumptions and the algorithm. The proposed TopRank algorithm is provable to have a weaker modeling assumption, which I believe is an important contribution. The theoretical guarantees and empirical justifications on real data are companioned which seem to be technically sound. I think it is better to have a more detailed discussion on Assumption 4 since it is relative unnatural comparing with the other three assumptions. Explanations of its necessity from the algorithmic and theoretical aspects, and the potential way of relaxation are desirable. My major concern is on the cost of storing the relational graph which records the preference of the users. The size of the corresponding matrix is L by L, which has a quadratic dependence on the number of items. This may limit the application of this approach on large-scale real-world ranking problems. Is there a possible way to have a relaxation on this requirement? I suggest to use different line shapes for different approaches in experimental results.

Reviewer 3



The proposed method proposed an online learning algorithm of rank based on topological sort. It is an interesting idea of using partition in the learning of rank. But I would suggest the authors provide more explanation on why there is a need to use partition and what is the pros and cons of adopting such a strategy. In the current analysis, it lacks of analysis on computation complexity and effect on the number of blocks in the partition with respect to the performance on online learning. Some details comments: The current writing needs further improvement especially in the introduction section to make the key concept of the proposed method clearer such as the concept of “partial order” and “topological sort”. I would be better to have some simulation example to comprehensively evaluate the pros and cons of the proposed method, including computation complexity, estimation efficiency and accuracy.